# Towards fully integrated photonic displacement sensors

Ankan Bag [1,2,6], Martin Neugebauer[1,2,6], Uwe Mick[1,2], Silke Christiansen [1,3,4], Sebastian A. Schulz[5] & Peter Banzer[1,2 ✉]

The field of optical metrology with its high precision position, rotation and wavefront sensors represents the basis for lithography and high resolution microscopy. However, the on-chip integration—a task highly relevant for future nanotechnological devices—necessitates the reduction of the spatial footprint of sensing schemes by the deployment of novel concepts. A promising route towards this goal is predicated on the controllable directional emission of the fundamentally smallest emitters of light, i.e., dipoles, as an indicator. Here we realize an integrated displacement sensor based on the directional emission of Huygens dipoles excited in an individual dipolar antenna. The position of the antenna relative to the excitation field determines its directional coupling into a six-way crossing of photonic crystal waveguides. In our experimental study supported by theoretical calculations, we demonstrate the first prototype of an integrated displacement sensor with a standard deviation of the position accuracy below $\lambda/300$ at room temperature and ambient conditions.

[1] Max Planck Institute for the Science of Light, Staudtstr. 2, D-91058 Erlangen, Germany. [2] Institute of Optics, Information and Photonics, Department of Physics, Friedrich-Alexander-University Erlangen-Nuremberg, Staudtstr. 7/B2, D-91058 Erlangen, Germany. [3] Helmholtz-Zentrum Berlin für Materialien und Energie, Hahn-Meitner-Platz 1, D-14109 Berlin, Germany. [4] Physics Department, Freie Universität Berlin, Arnimallee 14, D-14195 Berlin, Germany. [5] SUPA, School of Physics and Astronomy, University of St Andrews, St Andrews, Scotland, UK. [6]These authors contributed equally: Ankan Bag, Martin Neugebauer. ✉email: peter.banzer@mpl.mpg.de

Optical metrology is one of the corner stones of quality control in modern nanotechnological devices, lithography and high accuracy localization. Grating based schemes and interferometers[1–3], both enabling deep sub-wavelength distance sensing for a huge variety of applications, are commonly utilized. Driven to the extreme, interferometers can even be used to sense gravitational waves, which requires a sensitivity to displacements on the order of $10^{-22}$ m, while requiring arm lengths on the order of km[4]. However, also in nano-optics and microscopy, localization accuracies in the deep sub-wavelength region can be reached[5,6], paving the way for super-resolution microscopy[7,8] and optical nanometrology[9].

Recently a technique for two-dimensional localization based on position-dependent directional scattering of a single dipolar nanoantenna has been demonstrated[10–12]. This transverse Kerker scattering of a nanoantenna approach is based on the tailored excitation of so-called Huygens dipoles[13,14], combinations of interfering electric and magnetic dipoles that result in the conceptually highest directivity possible for single dipolar emitters[15,16].

Here we implement this scheme to measure the relative position between an external light source and a photonic microchip. For excitation, a tightly focused radially polarized beam impinges onto a silicon antenna accurately placed in the center of a six-way crossing of a 2-D photonic crystal waveguides (PCW), formed by removing single rows of holes[17]. Depending on the position of the antenna relative to the cylindrically symmetric three-dimensional electromagnetic field landscape of the excitation beam, the light is coupled into the individual waveguide arms of the device with a higher or a lower efficiency. In our experiment, we detect the amount of light coupled into each arm by imaging six out-couplers from the far field, while future devices will also include integrated detectors. The strong position dependence of the directional coupling allows for a straight forward implementation of our device as two-dimensional displacement sensor that extends the nanometrological toolbox.

## Results

**Theoretical concept**. First, we elaborate on the general theoretical concept and consider an incoming tightly focused radially polarized beam propagating in the $z$-direction[18,19]. The beam has a width $w_0 = \mathrm{NA}f$ before focusing, where $f$ is the focal length of the focusing lens and NA is its numerical aperture (as an example we use NA = 0.9). In the focal plane ($x$–$y$ plane), the cylindrically symmetric beam carries a strong longitudinal electric field component ($E_z$) in the center (on the optical axis of the lens), surrounded by an azimuthally polarized magnetic field ($H_\phi$) and a radially polarized electric field ($E_r$)[20]. Plots of the corresponding energy densities $w_E^z$, $w_H^\phi$, and $w_E^r$ are shown in Fig. 1a, calculated using vectorial diffraction theory[21,22].

All other field components, $E_\phi$, $H_r$, and $H_z$, are equal to zero for symmetry reasons. The wavelength $\lambda$ in comparison to the size of the focal spot is indicated by the black bar in the upper left panel. The relative phases of the individual field components are plotted as insets. It is important to note that a relative phase of $\pm\pi/2$ between longitudinal and transverse fields is observed. If a dipolar Mie-scatterer is placed in such a field distribution, electric and magnetic dipole moments will be excited proportional to the local field vectors[23,24], $\mathbf{p} \propto \mathbf{E}(\mathbf{r})$ and $\mathbf{m} \propto \mathbf{H}(\mathbf{r})$. Therefore, longitudinal electric and transverse magnetic dipole moments are excited simultaneously whenever the antenna is at a position where the corresponding fields overlap. As mentioned before, our approach is based on so-called Huygens' dipoles that are excited whenever the particle moves away from the optical axis. In general, the electric and magnetic dipole emissions have to be

in-phase for a Huygens' dipole to realize maximum directional interference[10,11,14,25,26]. With our choice of transverse and longitudinal field components exciting the particle, a lateral directivity (transverse Kerker scattering) can be achieved, enabling position-selective coupling to the waveguide architecture discussed below. To compensate for the relative phase between longitudinal and transverse field components in the focus[24], the phase relation between the particle's lowest-order Mie-coefficients[11] can be exploited by choosing an appropriate wavelength. Owing to the cylindrical symmetry of the excitation field, the direction of lateral scattering and coupling depends on the azimuthal position of the particle, while the strengths of the directionality is given by the radial distance of the particle relative to the center of the beam[11,24]. In other words, the position of the particle can be determined by the directionality of the scattered light. For illustration of the concept, we marked two positions in each panel of Fig. 1a. For an on-axis position (marked with a cross) only an electric $z$-dipole can be excited[20]. The resulting symmetric far-field emission pattern is sketched in real space for the $x$–$y$ plane in Fig. 1b, with the calculated angular spectrum (AS) intensity distribution depicted below, where we use the $k$-space coordinates $k_x$ and $k_y$ corresponding to $x$ and $y$ in real space. The black dotted circles mark the transition from propagating waves (plane waves that propagate into the far field when the dipole is situated in free-space) with $k_\perp \equiv (k_x^2 + k_y^2)^{1/2} \le k_0 \equiv 2\pi/\lambda$, to the evanescent part of the AS[22,27,28], $k_\perp > k_0$. However, when the particle is displaced, an additional magnetic dipole moment will be excited. As an example we consider a displacement along the $x$-axis (see black squares in Fig. 1a) which results in a magnetic $y$-dipole moment. Note here that an electric $x$-dipole moment is excited as well. However, its influence on the AS can be neglected for the present configuration, since magnetic $y$-dipole moment and the electric $z$-dipole moment dominate the scattering behavior in the region close to the optical axis[11]. The optimal case, when the electric $z$-dipole and the magnetic $y$-dipole exhibit the same strength and are in-phase (Huygens dipole[13,14]) results in the strongly directional AS depicted in Fig. 1c. Most importantly, this directionality occurs not only in the propagating part of the AS—implying transverse Kerker scattering to the far field as indicated by the sketch—but also in the evanescent part[14,26]. This means that the Huygens dipole will also couple directionally to a close-by waveguide[14]. For the PCW architecture utilized in our device described below, we estimate the coupling to occur for the six transverse $k$-vectors marked by the black triangles in Fig. 1b, c. For the example plotted in Fig. 1c, we therefore expect to observe more coupling into the directions corresponding to $k_x < 0$, in particular along the axis with $k_y = 0$.

Consequently, by measuring the directivity coupled to the six arms of our device, it is possible to obtain information on the position of the antenna relative to the optical axis of the beam. In the following we therefore discuss an experimental prototype implementation of the approach as a two-dimensional displacement sensor.

**Implementation**. An electron micrograph of the prototype device is depicted in Fig. 2a.

It comprises an under-etched hexagonal photonic crystal lattice (lattice period $a = 424$ nm, hole diameter $d = 0.55a$, slab thickness $t = 220$ nm) including three PCWs (see lines of missing holes in Fig. 2b) along 0°, 60°, and 120°. A spherical silicon nanoparticle is placed at the six-way crossing point of the waveguides (see magnified image of the crossing point in Fig. 2c) utilizing a pick-and-place handling approach[29]. The particle acts as an optical antenna that, when excited by an incoming beam,

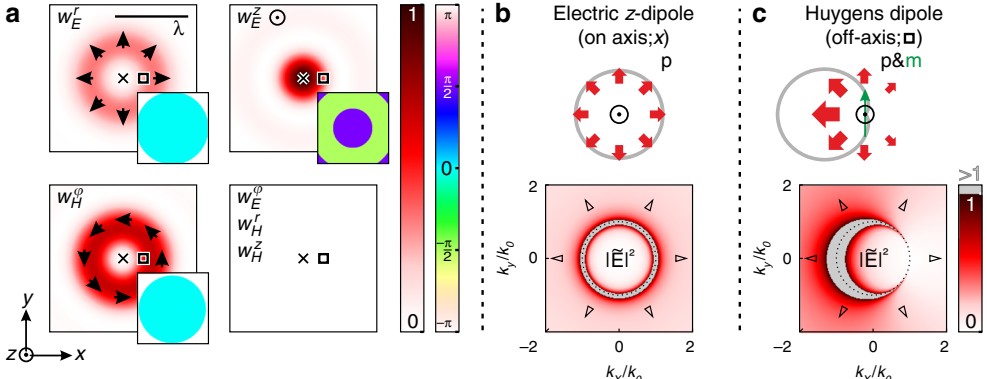

**Fig. 1 Position-dependent directionality. a** Electromagnetic field distribution of a tightly focused radially polarized beam. The focal energy density distributions of the radial electric, $w_E^r = \frac{\epsilon_0}{4}|E_r|^2$ (upper left), the longitudinal electric, $w_E^z = \frac{\epsilon_0}{4}|E_z|^2$ (upper right) and the azimuthal magnetic $w_H^\phi = \frac{\mu_0}{4}\left|H_\phi\right|^2$ (lower left) components are shown. The distributions of $w_E^\phi$, $w_H^r$, and $w_H^z$ are strictly zero everywhere (lower right panel). The black arrows indicate the local orientation of the corresponding fields, while the relative phases are shown as insets. The crosses and the squares in each panel mark two especific positions within the beam. **b** Angular spectrum (AS) electric field intensity distribution $|\tilde{\mathbf{E}}|^2$ of an electric dipole oriented in $z$-direction (expected at the position marked by the crosses in **a**). **c** The distribution of $|\tilde{\mathbf{E}}|^2$ for a combination of an electric dipole oriented in $z$-direction and a magnetic dipole oriented in $y$-direction (expected at the position marked by the square in **a**). At the boundary between propagating and evanescent parts of the AS (see dotted black circles), $|\tilde{\mathbf{E}}|^2$ exhibits a singularity. The color-code is scaled in order to achieve a good visibility in the region of interest, while higher values of $|\tilde{\mathbf{E}}|^2$ (>1 in the chosen color-map) are not considered (see gray areas). The six black triangles indicate the $k$-vectors for which the light is estimated to couple to the six arms of the photonic crystal structure described below.

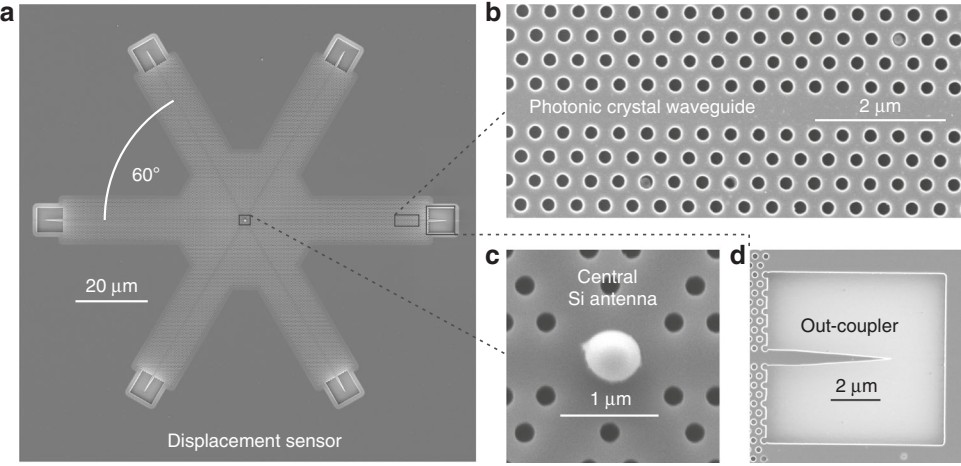

**Fig. 2 Design of the photonic displacement sensor. a** The electron micrograph shows the complete structure consisting of a six-way photonic crystal waveguide crossing (see waveguide section in **b**), a Si antenna with radius $r \approx 260$ nm placed in the center of the crossing (depicted in **c**), and six tapered out-couplers (see example in **d**).

couples light into the respective arms of the PCWs. At the end of each arm the light is coupled out of the PCWs (see magnified image of the tapered coupler in Fig. 2d) and scattered into the far field where it can be measured[30]. In future applications, the out-couplers can be replaced by integrated detectors[31,32]. More details on the fabrication of the sample can be found in the Methods section.

For the discussion of the optical properties of the device we consider a monochromatic excitation field with wavelength $\lambda = 1608$ nm, which is within the band gap corresponding to the fundamental transverse magnetic mode of the PCWs. The size of the central antenna (radius $r \approx 260$ nm) is chosen such that the electric and magentic Mie-coefficients for the given wavelength are compensating for the relative phase of the field components in the beam[11]. In general, the optimal particle size can be estimated for a given wavelength using Mie-theory[10]. For the particle in the experiment we obtain a ratio between the magnetic and electric polarizabilities ($\alpha_m$ and $\alpha_e$) of $\alpha_m/\alpha_e \approx 0.63e^{i0.58\pi}$. The phase

difference between $\alpha_m$ and $\alpha_e$ is close to the optimum ($\pi/2$), which means it should be possible to couple to the individual waveguide arms of the structure with the strong directionality of a Huygens dipole[14], although the amplitude ratio below 1 leaves room for optimization for future devices[10].

The experimental setup used for testing the displacement sensor is depicted in Fig. 3a. The incoming radially polarized beam of light—radial polarization provided by a $q$-plate (not shown here) with charge 1/2[33]—passes through a non-polarizing beam-splitter and is subsequently tightly focused by a high NA (NA = 0.9) microscope objective (MO) onto the sample. Using a three-axis piezo stage, the antenna in the center of the structure can be positioned precisely with respect to the incoming excitation field. The amount of light coupled to the individual arms of the waveguide is measured by imaging the sample onto an InGaAs camera, utilizing the same MO and an additional lens. An exemplary optical false color image of the displacement sensor superimposed on an electron micrograph of the device can be

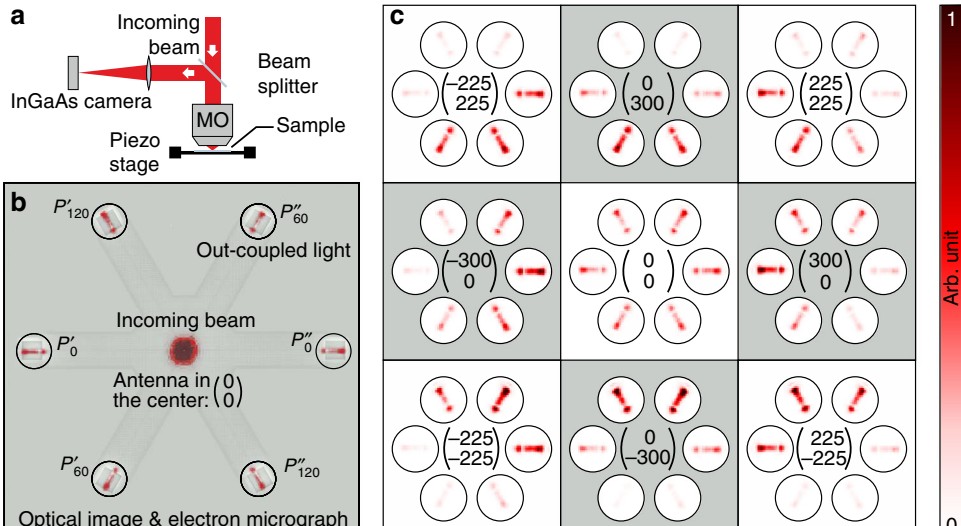

**Fig. 3 Optical setup and directional coupling. a** The simplified sketch of the experimental setup shows an incoming radially polarized beam passing through a beam-splitter and being tightly focused by a high numerical-aperture microscope objective (MO) onto the displacement sensor. The sample is imaged by the same MO and an additional lens onto an Indium gallium arsenide (InGaAs) camera. **b** The exemplary optical image of the sample superimposed on an electron micrograph of the structure shows the incoming beam (see overexposed spot in the center) and the six out-couplers for the antenna being positioned close to the center (optical axis) of the beam. **c** The images of the out-couplers are plotted for nine different positions relative to the beam, with the x- and y-coordinates being denoted in units of nm in the center of each panel.

seen in Fig. 3b, where the antenna is approximately in the center of the incoming beam, with $(x, y) = (0, 0)$. The overexposed spot in the center corresponds to the incoming beam reflected by the sample. Besides this central spot, only the out-couplers highlighted by six black circles scatter light to the far field, demonstrating the quality of the PCWs. As expected for an electric z-dipole moment being excited in the antenna, all out-couplers can be observed with a similar visibility, indicating that the light is coupled symmetrically to the individual arms of the device (compare with Fig. 1b).

As a first indicator for the functionality of our prototype displacement sensor we depict the out-couplers for 8 additional positions of the antenna relative to the beam in Fig. 3c. These positions are denoted by the vectors $(x, y)$ in each panel. As we can see, the displacement of the sample results in directional coupling. For example a displacement in positive or negative x-direction leads to directional coupling to the left or to the right, respectively. In particular, at the position $x = 300$ nm, $y = 0$, the measured coupling efficiencies resemble the AS of the Huygens dipole at the positions in k-space marked by the six black triangles in Fig. 1c. Hence, the measurement verifies the theoretical concept of directional waveguide coupling of Huygens dipoles[14]. To the best of our knowledge, this is the first experimental application of a Huygens dipole for directional coupling to PCWs. In addition, this effect can also be observed for displacements along the y-axis, the $+45°$-axis, and the $-45°$-axis (see Fig. 3c). This demonstrates the tunability of the coupling direction by position-dependent structured illumination[34].

In order to provide further validation of the measurement results, we theoretically simulate our system with the finite-difference-time-domain method where we use the same waveguide, particle, and beam parameters as in the experiment. The geometry is shown in the top-view and side-view sketches in Fig. 4a.

The color-coded distributions in Fig. 4b, c show the total energy density of the electric field, $w_E = \frac{\epsilon_0}{4}|\mathbf{E}|^2$, for a plane of observation in the center of the waveguide (see dashed blue line in the side-view sketch in Fig. 4a). Both distributions are normalized

to the same value. When the incoming beam impinges centrally on the structure, we observe symmetric coupling to the six arms of the waveguide. However, shifting the beam with respect to the sample (we consider a displacement of 300 nm along the x-axis) results in strongly asymmetric coupling. Both results match the experiments shown in the corresponding panels of Fig. 3c. In order to provide a quantitative comparison of the directional coupling we determine the ratio between the strong (left) and the weak (right) coupling for the 0° (horizontal) waveguide. The numerical simulation shown in Fig. 4c results in a ratio of 6.6. In the experiment, we achieved a lower value of 4.5 which is calculated as the average coupling ratio for the displacements along the x-axis of $\pm300$ nm shown in Fig. 3c. The mismatch can be attributed to aberrations of the incoming beam, small deviations between the experimental sample and the ideal geometry of the simulation, and back reflections at the tapered out-couplers.

We conclude that our prototype displacement sensing device couples light directionally to the individual arms of the waveguide, depending on the location of the antenna within the excitation field.

**Quantitative analysis**. Finally, we perform a quantitative analysis of the displacement sensing capabilities of the device. We begin with a calibration measurement where we raster scan the particle across the central region of the beam with a step size of 10 nm and a total number of $21 \times 21$ steps. For each position set by the piezo stage $(x, y)$ we take several images of the sample, determine the power emitted by each out-coupler (sum over the measured intensity values within the six black circles as exemplarily shown in Fig. 3b), and evaluate the directivity parameters,

$$D_i = \frac{P_i'' - P_i'}{P_i'' + P_i'}, \qquad (1)$$

for all three waveguides ($i = 0, 60, 120$). The results are summarized by the plot of the position-dependent directivities $D_0$ (red dots), $D_{60}$ (green dots), and $D_{120}$ (blue dots) in Fig. 5a.

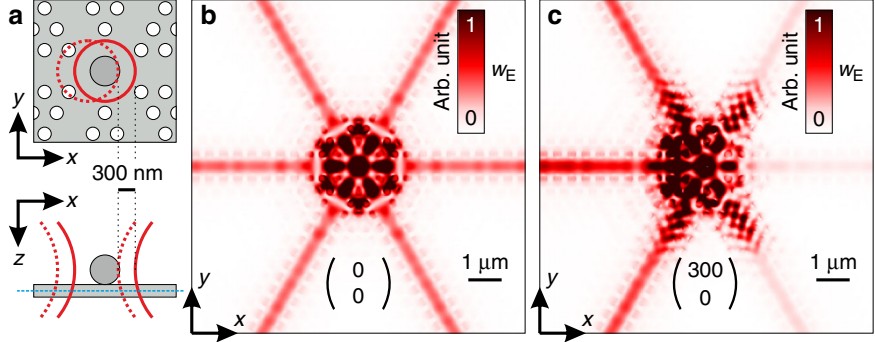

**Fig. 4 Finite-difference-time-domain simulations of the position-dependent waveguide coupling. a** Top-view (upper panel) and side-view (lower panel) sketch of the simulation geometry. The Si antenna is shown in dark gray and the PCW crossing in light gray. The excitation beam is indicated by solid (beam in the center) or dashed (beam shifted by 300 nm) red lines. The plane of observation is marked by the dashed blue line in the side-view sketch. **b**, **c** Numerically calculated distributions of the total electric field energy density $w_E$ for the beam being in the center and the beam being shifted with respect to the sample by 300 nm along the $x$-axis.

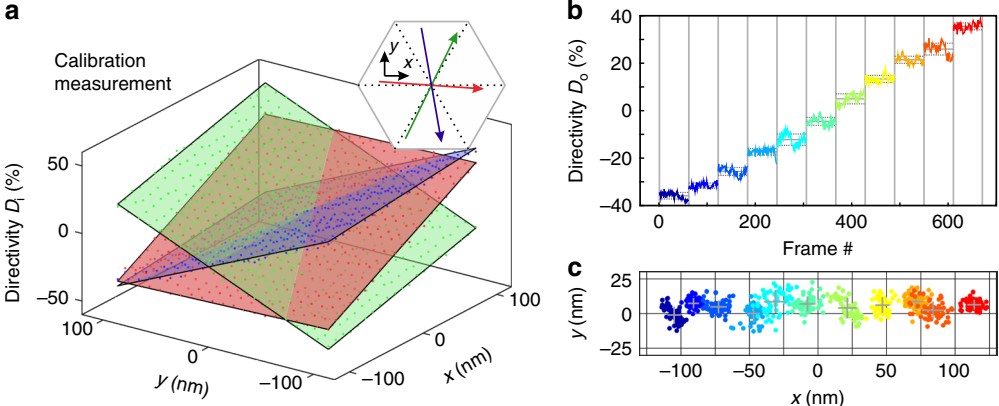

**Fig. 5 Calibration measurement and experimental localization results. a** The directivity parameters $D_0$ (red dots), $D_{60}$ (green dots), and $D_{120}$ (blue dots) are plotted against the corresponding positions set by the piezo stage. The semi-transparent color-matching planes correspond to two-dimensional linear equations that are fitted to the experimental data. The gradient vectors of the individual planes are plotted as arrows in the hexagonal inset, while the actual orientations of the waveguides are indicated as black dotted lines. **b** The directivity $D_0$ is plotted for a line scan along the $x$-axis with a step size of 25 nm. For each position (denoted by one single color) set by the piezo stage, 61 camera frames are captured and the directivity parameters are determined. **c** For the same scan, the two-dimensional position is calculated using the calibration data.

For each waveguide we fit a set of two linear equations to the experimental data (see Methods section), which are indicated as red ($D_0$), green ($D_{60}$), and blue ($D_{120}$) semi-transparent planes. The good overlap between the experimental data and the fitted planar equations validates the calibration process and the assumption of linearity[11,35]. Especially the gradients of the three calibration planes can be used to assess the quality of our displacement sensor. For this reason we plot the gradients of each plane as red, green, and blue arrows in the upper right inset. The vectors basically follow the orientations of the waveguides (dotted black lines), with the biggest angular mismatch of ≈20° between the blue arrow and its corresponding waveguide, which can be attributed to aberrations of the incoming beam and the imperfections of the sample. The amplitudes of the gradient vectors represent an important measure of the sensitivity $S$ of our device, which is defined as change in directivity per nanometer displacement[10,11,36]. Here we achieve sensitivities of $S_0 \approx 0.33\%$ nm$^{-1}$, $S_{60} \approx 0.37\%$ nm$^{-1}$, and $S_{120} \approx 0.37\%$ nm$^{-1}$ that are up to one order of magnitude lower than recently reported comparable experimental results[10,11,37], which were achieved in the visible regime and without integration on a photonic chip. However, depending on the actual detector system, signals on the order of 1% and below are measurable, implying that displacements in the nanometer regime are observable.

Finally, in order to experimentally estimate the resolution of our integrated displacement sensing prototype, we perform a line scan along the $x$-direction—11 positions with a step size of 25 nm each—where we measure 61 images for each position set by the piezo stage. Calculating the directivity parameter $D_0$ (the most sensitive parameter for shifts along the $x$-axis) for all camera frames results in the step function shown in Fig. 5b, where each color-coded step corresponds to a different position. While the individual 25-nm steps can be observed with ease, we also see fluctuations of $D_0$ within each step, which are caused by real jittering of the sample relative to the beam and the noise of the camera. The measured standard deviations of $D_0$ of each step are indicated as horizontal, dotted black lines, while the median is indicated as solid line. The average standard deviation is ≈1.7%, corresponding to a positioning accuracy of ≈±6 nm.

As a next step we use the calibration measurement to calculate the actual position of our sample relative to the beam (see Methods section for details). The results are plotted as colored dots in Fig. 5c. The color-code of the individual measured positions corresponds to the color-code of the individual steps in Fig. 5b. Again we can observe the stepwise trajectory of the sample along the $x$-axis, since the measured positions are mostly separated. Along the $y$-axis we only observe fluctuations. As an estimate for the resolution of the position measurement we

additionally plot the average position (centroids of the gray crosses) and the corresponding standard deviations in positive and negative x- and y-directions (extent of the crosses in the corresponding directions), which are of the order of ±5 nm, respectively.

This estimate of the position accuracy, however, represents only an upper bound for the actually achievable localization accuracy, since the relatively high standard deviations are partially caused by actual variations of the particle position relative to the beam as a result of vibrations and drifts in the setup.

## Discussion

In this article we demonstrated a prototype implementation of an integrated nanophotonic displacement sensor with a resolution in the nanometer range. The device is based on position-dependent directional coupling of a Huygens dipole to a six-way photonic crystal waveguide crossing. The Huygens dipole moment is induced in a silicon antenna by a spatially varying, cylindrically symmetric excitation field. To the best of our knowledge, this is the first experimental demonstration of controllable nanoscale light routing by a Huygens dipole in a PCW platform.

Therefore, the presented experiments represent an important step toward fully integrated optical devices even beyond displacement sensing. More complex designs might be implemented to build chip-scale combined wavelength[38], polarization[39–41], position[10–12,42], and wavefront (tilt)[43] sensors. The realization of an integrated device, capable of sensing multiple degrees of freedom of an external light source simultaneously, might become relevant for applications like sample stabilization in microscopy[6], adaptive optics, and acceleration sensors.

## Methods

**Angular spectrum of a Huygens dipole**. The angular spectra (AS) of arbitrarily oriented electric and magnetic dipoles can be written as[44]

$$\tilde{\mathbf{E}}_{\mathrm{ED}}\left(k_x, k_y\right) = C\left[(\mathbf{e}_s \cdot \mathbf{p})\mathbf{e}_s + \left(\mathbf{e}_p \cdot \mathbf{p}\right)\mathbf{e}_p\right], \tag{2}$$

$$\tilde{\mathbf{E}}_{\mathrm{MD}}\left(k_x, k_y\right) = C\left[\left(\mathbf{e}_s \cdot \frac{\mathbf{m}}{c}\right)\mathbf{e}_p - \left(\mathbf{e}_p \cdot \frac{\mathbf{m}}{c}\right)\mathbf{e}_s\right], \tag{3}$$

$$C = \frac{\imath k_0^2}{8\pi^2 \epsilon k_z}, \tag{4}$$

with the electric permitivity $\epsilon$, the speed of light $c$, the longitudinal component of the $k$-vector, $k_z = (k_0^2 - k_\perp^2)^{1/2}$, and the unit vectors along $s$- and $p$-polarization basis, $\mathbf{e}_s$ and $\mathbf{e}_p$. Here we consider a combination of an electric dipole oriented in $z$-direction ($p_z$) and a magnetic dipole oriented in $y$-direction ($m_y$). The total AS is the sum of $\tilde{\mathbf{E}}_{\mathrm{ED}}$ and $\tilde{\mathbf{E}}_{\mathrm{MD}}$, which results in

$$\tilde{\mathbf{E}}\left(k_x, k_y\right) = C\left[\left(\frac{k_x k}{k_z k_\perp}\frac{m_y}{c} - \frac{k_\perp}{k_z}p_z\right)\mathbf{e}_p - \frac{k_y}{k_\perp}\frac{m_y}{c}\mathbf{e}_s\right]. \tag{5}$$

The distribution of the angular spectrum intensity $|\tilde{\mathbf{E}}|^2$ in Fig. 1b is calculated for $m_y = 0$. The distribution in Fig. 1c corresponds to the Huygens dipole with $m_y/c = p_z$. At the critical angle ($k_\perp = k_0$), $k_z = 0$, which results in $|\tilde{\mathbf{E}}|^2 \to \infty$. Therefore, the regions close to said angle, where the intensity surpasses a certain maximum value, are not displayed in Fig. 1b, c as indicated by the gray areas. This enhances the visibility in the regions of interest.

**Sample preparation**. The sample is based on the silicon-on-insulator technology, with a 220-nm silicon top layer on a 2000-nm oxide layer. The PCW and tapered coupler pattern was defined in a ZEP520A resist layer using a Raith E-line 30 kV electron beam lithography system. Following resist development, the pattern was transferred into the silicon layer using a SF6/CHF3 reactive ion etch. A symmetric membrane was achieved by removing the oxide underneath the taper and PCW region using an HF etch.

The PCW parameters ($a = 424$ nm, hole diameter of 233 nm) are chosen such that there is a guided TM mode in the PCW at the desired wavelength.

The out-coupling interface is designed specifically for this position sensor, to provide a robust, wavelength independent coupling that can be detected using the same lens used for the excitation, i.e., requiring coupling in the normal direction, with low back reflections. The free-standing taper results in a low-reflection

interface. The distance between the taper tip and the silicon edge is chosen such as to maximize the upward scattering efficiency. This simple taper provides ~33% upward scattering efficiency with <2% back reflections. The low back-reflection is critical to be able to properly observe the directivity parameters.

The silicon particle with a diameter of ~520 nm, acting as a dipole antenna, was transferred to the waveguide crossing by an AFM-based nanomanipulation setup. The system is operated inside a scanning electron microscope (SEM). Van-der-Waals forces enable picking up and placing the desired nanoparticles. Since the target sample was tilted in the setup to observe the transfer of the particle, the main inaccuracies for placing the particle at the desired target location emerge from the limited estimation of the particle's location on the waveguide crossing along the SEM's line of sight (along the sample's y-direction). Nevertheless, it was possible to place the silicon particle with an offset of only 20 nm from its desired target position at the center of the waveguide crossing. After placing the silicon nanoparticle on the photonic crystal waveguides, it is held in position through Van-der-Waals forces and the sample is transferred from the SEM/AFM to the experimental setup described in the main text.

**Calibration measurement**. In close vicinity to the optical axis of the incoming beam, the directivity parameters summarized by the vector $\mathbf{D} = (D_0, D_{60}, D_{120})$ are linearly dependent on the antenna position $\mathbf{r} = (x, y)$. Therefore, we can fit a system of linear equations to the averaged calibration measurement data,

$$\mathbf{D} = \hat{\mathbf{M}}\mathbf{r} + \mathbf{O}, \tag{6}$$

where the matrix $\hat{\mathbf{M}}$ and the vector $\mathbf{O}$ entail the gradients and the offsets of the calibration curves:

$$\hat{\mathbf{M}} = \begin{pmatrix} 0.33\frac{\%}{\mathrm{nm}} & -0.02\frac{\%}{\mathrm{nm}} \\ 0.16\frac{\%}{\mathrm{nm}} & 0.33\frac{\%}{\mathrm{nm}} \\ 0.06\frac{\%}{\mathrm{nm}} & -0.36\frac{\%}{\mathrm{nm}} \end{pmatrix}, \mathbf{O} = \begin{pmatrix} -1.75\% \\ 2.54\% \\ 1.85\% \end{pmatrix}. \tag{7}$$

The 95% confidence bounds of the matrix elements and of the components of the offset vector are $\approx \pm 4.5 \times 10^{-3}\%\,\mathrm{nm}^{-1}$ and $\approx \pm 0.28\%$, respectively.

A measurement of all three directivity parameters results in an overdetermined system, which can be solved for $x$ and $y$ by using the pseudo inverse:

$$\mathbf{r} = \left(\hat{\mathbf{M}}^T \hat{\mathbf{M}}\right)^{-1} \hat{\mathbf{M}}^T (\mathbf{D} - \mathbf{O}). \tag{8}$$

This approach was utilized to calculate the sample positions depicted in Fig. 5c.

## Data availability

The datasets generated during the current study, which supports the paper, may be obtained from the digital object identifier (DOI)[45].

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

## Acknowledgements

The authors gratefully acknowledge the fruitful discussions with Gerd Leuchs. This project has received funding from the European Union's Horizon 2020 research and innovation program under the Future and Emerging Technologies Open grant agreement Super-pixels No. 829116. The authors thank James Burch and Alasdair Fikouras for helping S.A.S. with the fabrication of the waveguide architecture.

## Author contributions

P.B. conceived the idea and the experiment. A.B. performed the measurements and carried out the numerical simulations. M.N. performed the analytical calculations. A.B. and M.N. analyzed the experimental data. A.B. and S.A.S. designed the waveguide architecture. S.A.S. fabricated the waveguide architecture. U.M. and S.C. positioned the nanoparticle on the waveguide. M.N., A.B., and P.B. wrote the paper. P.B. supervised the project. All authors discussed the results and commented on the final paper.

## Competing interests

The authors declare no competing interests.
