## [Peer Review File · Nature Communications]

Reviewers' comments:

Reviewer #1 (Remarks to the Author):

This was a beautifully written manuscript and was a pleasure to read and review. The authors build upon their previous work on the on nanoparticle localisation using the Kerker condition (Ref. 10) and implement an integrated displacement sensor on a photonic crystal waveguide. The theoretical basis is explained well in a fashion that is accessible even to non-experts and the experimental realisation is clear and convincing. I believe that the results are significant and interesting to a wide readership, and I recommend the publication provided the following concerns are addressed.

1. It is approximated that the electric dipole moment induced in the particle is proportional to the local electric field and the magnetic dipole moment is proportional to the local magnetic field (and presumably, the other moments are insignificant). How reasonable is this approximation given that the diameter is 520 nm for a wavelength of 1608 nm considering the high refractive index of Silicon? Performing a multipole decomposition of the fields on the sphere would be convincing.

2. The usage "Estimated position" in Page 3 and the caption of Page 4 is confusing. Is the position being read out directly from the piezo, or is it estimated from the experimental result? Please clarify.

3. The 20° mismatch of the blue arrow from its corresponding waveguide in Fig. 5a is quite significant. The authors attribute it to "aberrations of the incoming beam and the imperfections of the sample", but this needs further elaboration. If the issue is with the incident beam, it should be possible to show it by rotating the sample/beam. The methods section (VI B) talks about a possible offset in the placement of the central particle by about 20 nm. Could such a mismatch explain the difference? (If so, a simulation like in Fig. 4 should be able to show it qualitatively).

4. My major concern is regarding the error analysis in the manuscript. I believe that a few additional details are required to make the claims of accuracy:

A. That the directivity vector changes linearly with displacement is only a first-order approximation valid around the centre. How far from the centre can the particle be displaced before the higher-order terms become significant? Plotting the deviation of the points in Fig. 5a from the respective planes should make it clear whether the 100nm x 100nm region remains within the linear regime. (Arguably, this is not a serious concern because a non-linear fit could be used to invert the function. But for the correctness of the analysis in the manuscript, it matters).

B. The error estimate is only shown for displacement along x. Does the same value of error hold for orthogonal displacement? This is significant because y-displacement will not be along one of the waveguide directions, and perhaps the gradient offset discussed above plays a role as well.

C. The manuscript takes the standard deviation of the multiple measurements to be the error of measurement. But this only gives us an estimate of the precision of the experiment, not of the accuracy. What needs to be shown is how well the extracted value matches with the actual displacement from the piezo (modulo a global shift, say). For instance, I did a rough measurement of the horizontal separation between centroids from Fig. 5c and found that the shifts varied from 12nm to 29 nm. The horizontal separation between the first and last point is only 216 nm instead of the expected 250 nm. This points towards much larger inaccuracies than claimed, perhaps systematic. Overall, based on these numbers, my feeling is that the localisation accuracy might even be as high as 15-20 nm than the claimed 5-6 nm.

A two-dimensional map of the offset between actual position (from the piezo) and the estimated position from the experiment will clear these issues and provide a more robust estimate of the error. I understand that this is an elaborate experiment to perform and repeating might not be an option given the time that has passed. But at the very least, please perform this analysis using the data points already in Fig. 5a to clarify points 4B, C.

Minor additional comments

1. The manuscript consistently uses the term "Huygens dipole", but many in the dielectric community might be more familiar with the terminology of Kerker conditions. Please specify this too in the manuscript.
2. In the caption of Figure 3, the subfigures have been listed incorrectly as (c,d) instead of (b,c).
3. Please re-check the equations in section VI A. Comparing with Ref. 41, I found the following issues:
 - A. Dot products ($\mathbf{e}_s \cdot \mathbf{p}$ etc.) are missing in Eq. 2
 - B. \mathbf{e}_s and \mathbf{e}_p are not defined
 - C. There is a mismatch in the prefactor C, it seems to differ by k_0/k_z

Reviewer #2 (Remarks to the Author):

The authors present a position sensing detector proof-of-concept based on Huygens dipole scattering of radially polarised light on a Si nanoparticle. The directionally scattered light is picked up and channeled to a output-coupler by means of six photonic crystal waveguide. The authors demonstrate very high lateral resolution, which is order of magnitude better than what is typically achievable with commercial position sensing detectors. The work successfully presents a new and very interesting concept with a huge potential for many applications. I find the manuscript ready for publication as is (with a single typo in the subfigure naming in the caption of Fig. 3).

Reviewer #3 (Remarks to the Author):

This is a very interesting research work putting forward a novel method for high precision control of position using nanoparticles that may be modelled as emitting Huygens dipoles.

The manuscript is generally well written, and makes use of recent important concepts developed in the field of nanophotonics. I particularly like the details given in the Quantitative Analysis section.

I recommend its publication in Nat. Comm., only having a few comments that the authors might consider to improve it.

1. Incoming is not the same as incident. An incident source-free field, (e.g. a plane wave or beam) may have both incoming and outgoing waves. ¿Is the illumination of the sample an incoming spherical wave converging to the spot where the object (i.e. the particle) is? Or is rather an incident wavefield?.

2. Although it is later understood, perhaps the authors could explicitly state from the start that the photonic cristal is a 2-D one formed by long cylinders.

3. I had no clear how the particle was maintained in its position. The authors say they use an AFM to pick it up and place it at the spot of interest, but how is the particle kept there?. Is the AFM working while the particle is emitting at that point?. Or, are there holes in the junction of the arms where the particle may be left?. The authors explain the resolution of their set-up, but how are the fuctuations of the particle, jittering around the spot of interest, tighted to such a small displacement values?. Some details would help.

4.Connecting the above, is there no radiation pressure, gradient optical force, or photophoretic force, due to the infrared incident illumination that might hinder the stability of the particle position?.

5. I feel that it would be helpful to know beforehand where from does the illumination come. The

authors discuss right from the introduction the interplay in the creation of the e and m dipoles, but the reader has to wait seeing Fig. 3 a to realize that illumination comes from above.

6. I had difficulties to interpret the superíndices of P: prime and double prime in Eq.(1). Perhaps they could be made larger in Fig. 3 b.

7. The reference to the angular spectrum representation (ref. 20 in 24th line of left column in page 3) does not acknowledge the real sources, which is not Novotny's textbook, although he also includes it in his book.

These sources are:

L. Mandel and E. Wolf, *Optical Coherence and Quantum Optics*, 1st ed. (Cambridge University Press, Cambridge, UK, 1995); M. Nieto-Vesperinas, *Scattering and Diffraction in Physical Optics*, 1st ed. (J. Wiley, New York, 1991); 2nd ed. (World Scientific, Singapore, 2006).

To Reviewer #1:

Comment 1 by Reviewer #1:

It is approximated that the electric dipole moment induced in the particle is proportional to the local electric field and the magnetic dipole moment is proportional to the local magnetic field (and presumably, the other moments are insignificant). How reasonable is this approximation given that the diameter is 520 nm for a wavelength of 1608 nm considering the high refractive index of Silicon? Performing a multipole decomposition of the fields on the sphere would be convincing.

Author's response:

We thank the Reviewer for this very valuable comment. For a detailed discussion of contributing multipolar moments, a multipole decomposition is essential, which we also performed but do not show in the manuscript. Actually, we tried to avoid the discussion in the text to keep the theory part easy-to-access. Even though the diameter of the particle is quite large (520 nm), the chosen wavelength (1608 nm) together with the high refractive index of silicon allow us to neglect higher order multipole and only consider the dominant dipolar electric and dipolar magnetic moments. A comparatively strong quadrupolar magnetic response would kick in at a shorter wavelength and dies off very quickly towards longer wavelengths. A numerical calculation following generalized Mie theory can be used to confirm the spectral response of the said particle, sitting on a silicon surface (see attached figure). The corresponding T-matrix-based approach has been discussed in several previous research works of our group, including Ref. 10, and of course others. To keep the theoretical explanation simpler as well as understandable to non-experts, we had kept the discussion focused on the dipolar response (which is predominant at the chosen wavelength).

As can be seen, at around 1608 nm, all higher order (including MQ) multipole contributions are negligible.

Comment 2 by Reviewer #1:

The usage "Estimated position" in Page 3 and the caption of Page 4 is confusing. Is the position being read out directly from the piezo, or is it estimated from the experimental result? Please clarify.

Author's response:

Thank you for pointing out this obvious misuse of the word 'estimated'. The positions (0,0), (-300,0), (300,0) etc. are exact locations set for the piezo. We have corrected it in the main manuscript. The real particle position might of course vary due to fluctuations, which we measure with the technique discussed in the manuscript.

Comment 3 by Reviewer #1:

The 20° mismatch of the blue arrow from its corresponding waveguide in Fig. 5a is quite significant. The authors attribute it to "aberrations of the incoming beam and the imperfections of the sample", but this needs further elaboration. If the issue is with the incident beam, it should be possible to show it by rotating the sample/beam. The methods section (VI B) talks about a possible offset in the placement of the central particle by about 20 nm. Could such a mismatch explain the difference? (If so, a simulation like in Fig. 4 should be able to show it qualitatively).

Author's response:

We thank the reviewer for commenting on this very important aspect of our experiment. To us it was also surprising in the beginning that for only one of the waveguides such an angular offset of the calibration plane and the actual waveguide orientation is observed. We believe that the observed discrepancy, which actually is not influence our presented data negatively, because it is considered in the calibration process, must be a combination of different minor imperfections. The beam itself is not perfect and shows minor deviations from the ideal focal field. In addition, the sample itself – i.e. the particle and the WG PhC system – are not perfect (but close to perfect). E.g. the particle is slightly displaced from the actual center of the crossing waveguides as indicated in the text and by the Reviewer. However, this displacement is not aligned with one of the directions defined by the three waveguides, which is confirmed by a careful analysis of the SEM images of the central area. Again, this displacement alone cannot explain the 20° angular mismatch of one of the calibration planes. Especially, because we measure signal difference of the two ends of each waveguide for the calibration, which should translate a displacement of the particle with respect to the perfect central position on the WG crossing to a change in offset of those fitted planes but no angular deviation. Last but not least, a slight deviation of the particle shape from a sphere could contribute to the experimental calibration maps in the above-mentioned way. A slight deformation can modify the directional scattering as the individual dipole contributes differently to the overall scattering when exposed to different excitation field directions, making the T-matrix components more or less sensitive to small displacements of the sample relative to the excitation field, eventually resulting in a tilted calibration plane. The relative orientation of the major axis of the ellipsoidal particle with respect to the WGs would be the defining parameter here. However, according to our analysis, the most important factor was played by the beam aberration. A slight deviation from the perfect radial polarization incoming beam, in terms of intensity or phase distribution, makes a huge impact on the scattering process. This is why we attributed the observed deviation of the blue gradient vector from the waveguide direction to “aberrations of the incoming beam and the imperfections of the sample”.

Comment 4 by Reviewer #1:

My major concern is regarding the error analysis in the manuscript. I believe that a few additional details are required to make the claims of accuracy:

- A. That the directivity vector changes linearly with displacement is only a first-order approximation valid around the centre. How far from the centre can the particle be displaced before the higher-order terms become significant? Plotting the deviation of the points in Fig. 5a from the respective planes should make it clear whether the 100nm x 100nm region remains within the linear regime. (Arguably, this is not a serious concern because a non-linear fit could be used to invert the function. But for the correctness of the analysis in the manuscript, it matters).
- B. The error estimate is only shown for displacement along x. Does the same value of error hold for orthogonal displacement? This is significant because y-displacement will not be along one of the waveguide directions, and perhaps the gradient offset discussed above plays a role as well.
- C. The manuscript takes the standard deviation of the multiple measurements to be the error of measurement. But this only gives us an estimate of the precision of the experiment, not of the accuracy. What needs to be shown is how well the extracted value matches with the actual displacement from the piezo (modulo a global shift, say). For instance, I did a rough measurement of the horizontal separation between centroids from Fig. 5c and found that the shifts varied from 12nm to 29 nm. The horizontal separation between the first and last point is only 216 nm instead of the expected 250 nm. This points towards much larger inaccuracies than claimed, perhaps systematic. Overall, based on these numbers, my feeling is that the localisation accuracy might even be as high as 15-20 nm than the claimed 5-6 nm.

A two-dimensional map of the offset between actual position (from the piezo) and the estimated position from the experiment will clear these issues and provide a more robust estimate of the error. I understand that this is an elaborate experiment to perform and repeating might not be an option given the time that has passed. But at the very least, please perform this analysis using the data points already in Fig. 5a to clarify points 4B, C.

Author's response:

Thank you very much for the detailed analysis of our manuscript and for bringing up this point regarding the error analysis, which is one of the most important aspects of all localization techniques in particular, or metrology schemes in general. First of all, there are many different ways of quantifying localization accuracies or precision. Because our scheme measures a

relative distance of particle with respect to the beam it is more sophisticated to draw a hard line between the two terms accuracy and precision. Furthermore, the position defined by the piezo stage itself only defines a central position around which the sample position can fluctuate.

To start with **point A**, it is true that linearity is given in a small area around the optical axis as also indicated in the text. In Fig. 5a, we plot the data from a square scan area of 200 nm x 200 nm around the optical axis (centre). Hence, the largest distance to the optical axis is equals $100 \text{ nm} * \sqrt{2}$. Still, the linear fit to the directivity planes leads to a directivity of $\sim 0.35\%/nm \pm 0.0045\%/nm$ (95% confidence interval) as already mentioned in ‘C. Calibration measurements’ under the Methods section. The confidence interval statistically represents the deviation of individual points from the fitted plane. In other words, the linear fit is still very much appropriate even for distances larger than 100 nm. We agree with Reviewer #1 that a non-linear fit would be appropriate when leaving the linear region. For our experiment, we actually chose a scan area of 400 nm x 400 nm around the centre, but only show a smaller region of 200 nm x 200 nm area in the manuscript, where the first order approximation holds true very well, so does the linear directivity assumption. Beyond this central area, as ratio of the longitudinal (E_z) and the transverse (H_y) field components decreases (following Fig. 1) nonlinearly, so does the directivity.

The answer to **point C** follows from a similar line of arguments. It is clear from Fig. 5c that the 11 sets of position data do not span a range of 250 nm owing to the effects of decreasing directivity (in a nonlinear fashion) beyond the central area. Coming to the question of ‘accuracy’ vs. ‘precision’, we are convinced that using the standard deviation as an accuracy of the measurement (error bar) is even underestimating the capabilities of our current experimental setup in contrast to the Reviewer’s claim of an increased accuracy. The directivity slopes of the fitted curve are between 0.33%/nm and 0.37%/nm. Given with camera we can sense a change in the signal of way less than 1%, we are sure that the actual accuracy value would be even less than 5-6 nm. Hence, with the number provide and a detailed explanation of how it is defined and derived, we give an upper bound for the error.

For the concern raised in **point B**, the same value of error holds for orthogonal displacement or displacement along any direction. If there is a line scan along y-axis, or along any direction, other than along the waveguides, to obtain the position data, the calibrated matrix would be used. And for the particular particle and the beam, the calibration takes care of all the aberrations of the system. In our case, we have the offset and the directivity matrix with uncertainties (95% confidence bound) given in Eqn. 7. The main point of showing Fig. 5b-c was to showcase the same experiment, but with a simple and easy to understand manner.

Minor additional comments by Reviewer #1

1. The manuscript consistently uses the term "Huygens dipole", but many in the dielectric community might be more familiar with the terminology of Kerker conditions. Please specify this too in the manuscript.
2. In the caption of Figure 3, the subfigures have been listed incorrectly as (c,d) instead of (b,c).
3. Please re-check the equations in section VI A. Comparing with Ref. 41, I found the following issues:
 - A. Dot products ($e_s \cdot p$ etc.) are missing in Eq. 2
 - B. e_s and e_p are not defined
 - C. There is a mismatch in the prefactor C, it seems to differ by k_0/k_z

Author’s response:

1. Thank you for the suggestion. We have now introduced ‘Kerker’ scattering / condition, keeping coherence with ‘Huygens’ dipole’ in a few places in the ‘Introduction’ and ‘Theory’ section to help the reader.
2. We have corrected the typo in the figure caption of Fig. 3
3. Thank you for noting the typo in Eqn. 2. We have corrected it. Furthermore, we now define e_s and e_p , and added k_0/k_z in the prefactor C. There was another small typo in Eqn. 3, which we have corrected as well. All mentioned typos and errors were actually only present in the equations noted in the manuscript but not in our calculations. Hence, Eqn. 5 was and still is fully correct. Same is true for all values resulting from calculations.

To Reviewer #2:

We thank the Reviewer for his/her valuable feedback. We are very happy to read that “*The work successfully presents a new and very interesting concept with a huge potential for many applications.*” We are also glad that the Reviewer finds our manuscript “*ready for publication as is*”, except for a single typo in one figure caption, which we have now corrected. Thank you very much.

To Reviewer #3:

Comment 1 by Reviewer #3:

Incoming is not the same as incident. An incident source-free field, (e.g. a plane wave or beam) may have both incoming and outgoing waves. Is the illumination of the sample an incoming spherical wave converging to the spot where the object (i.e. the particle) is? Or is rather an incident wavefield?

Author's response:

We thank you for your comment and apologize for the confusion. In our case, an aplanatic lens system creates a tightly focused light beam, which while propagating along the z-direction, converges on the object, i.e. the silicon particle. That is why we refer to it as incoming excitation beam, throughout the manuscript.

Comment 2 by Reviewer #3:

Although it is later understood, perhaps the authors could explicitly state from the start that the photonic crystal is a 2-D one formed by long cylinders.

Author's response:

We thank the Reviewer for raising this point. We didn't realize that this information could be beneficial for a reader but we fully agree now that it should be mentioned to also be more accessible to non-experts. We have now included a more detailed description while introducing the photonic crystal waveguide in the Introduction section. We had earlier mentioned that the photonic crystal waveguides are formed by removing a single row of holes, which essentially implies that the silicon based photonic crystal consists of cylindrical air-holes, which are arranged in a periodic fashion.

Comment 3 by Reviewer #3:

I had no clear how the particle was maintained in its position. The authors say they use an AFM to pick it up and place it at the spot of interest, but how is the particle kept there?. Is the AFM working while the particle is emitting at that point?. Or, are there holes in the junction of the arms where the particle may be left?. The authors explain the resolution of their set-up, but how are the fluctuations of the particle, jittering around the spot of interest, tightened to such a small displacement values?. Some details would help.

Author's response:

We thank the reviewer for this comment. Apparently, the part of the fabrication process concerned with the particle placement wasn't a hundred percent clear. In the following we describe the steps in more detail. As described in Methods Section B. Sample Preparation, we utilize a custom-made AFM based pick-and-place method, which allows us to precisely place the particle at the centre of the crossing waveguide structure, as shown in Fig. 2c. No additional hole is necessary at this position. After the placement, the adhesion force between the substrate and the particle (van-der-Waals force) is sufficiently strong to keep the particle in place and immobilizing it at its position. Subsequently, the sample (PhC architecture with particle and substrate) is removed from the vacuum chamber of the SEM where the manipulation (particle placement) was done. In a last step, we place the sample in our experimental setup and perform the optical measurements described in the manuscript. So, the actual experiment is not performed under an AFM. The adhesion forces are so strong that the particle even stays in place when flipping of the sample or under small mechanical vibrations.

Regarding the second part of this comment: the particle is not moving with respect to the PhC architecture. They are moving together. In this context, a relative motion of sample with respect to the focused field resulting from fluctuations might also arise from a 'motion' of the beam itself. We measure the relative position. Mechanical vibrations and thermal fluctuations of the piezo, the sample holder, and the sample may contribute to movement relative to the excitation beam. Our signal detection scheme based on the transverse Kerker scattering can resolve even tiny but fast (kHz frequency; exposure time = 1/1000 s) fluctuations. Such fluctuations are presented in Fig. 5b and c, where, for a fixed piezo position, fluctuations of the experimentally retrieved position are observed (see scattering of data points for different data-set colors (positions))

Comment 4 by Reviewer #3:

Connecting the above, is there no radiation pressure, gradient optical force, or photophoretic force, due to the infrared incident illumination that might hinder the stability of the particle position?

Author's response:

This is a very interesting and relevant point we and we thank the reviewer for bringing it up. Theoretically, when the incoming beam excites the particle (for instance by inducing a Kerker dipole), an effective force acts on the particle (magnetolectric or Kerker force) in addition to the radiation pressure and gradient optical force. However, in our situation, these effects are negligible small given the mass of the system (particle+Si membrane+substrate etc.) and the strength of the adhesive force. The laser power effectively interacting with the particle is on the order of tens of microwatts. However, the forces can be made visible as we showed recently in an experiment in an optical tweezers system.

Comment 5 by Reviewer #3:

I feel that it would be helpful to know beforehand where from does the illumination come. The authors discuss right from the introduction the interplay in the creation of the e and m dipoles, but the reader has to wait seeing Fig. 3 a to realize that illumination comes from above.

Author's response:

We appreciate the reviewer's feedback here and agree that it could be confusing to some readers. However, in the 'Theoretical Concept' section, we had avoided any particular details of our experimental setup and parameters to keep the explanation as general as possible. And that is why the particular information, i.e., illumination from above, is introduced in the 'Implementation' section only. However, to prevent any confusion, we have now included the z-axis in Fig. 1. Furthermore, we mentioned already in the first version in II. On page one that the beam propagates along the z-axis.

Comment 6 by Reviewer #3:

I had difficulties to interpret the superindices of P: prime and double prime in Eq.(1). Perhaps they could be made larger in Fig. 3 b.

Author's response:

Thank you very much. We have changed now the font size in Fig. 3 b.

Comment 7 by Reviewer #3:

The reference to the angular spectrum representation (ref. 20 in 24th line of left column in page 3) does not acknowledge the real sources, which is not Novotny's textbook, although he also includes it in his book.

These sources are:

L. Mandel and E. Wolf, *Optical Coherence and Quantum Optics*, 1st ed. (Cambridge University Press, Cambridge, UK, 1995); M. Nieto-Vesperinas, *Scattering and Diffraction in Physical Optics*, 1st ed. (J. Wiley, New York, 1991); 2nd ed. (World Scientific, Singapore, 2006).

Author's response:

Thank you very much for bringing this to our attention. We agree that the original sources should be mentioned too. We have included those in the list.

Reviewers' comments:

Reviewer #1 (Remarks to the Author):

I thank the authors for the detailed reply and making most of the corrections accordingly.

However, I am unsatisfied with the author response to my queries about error analysis. Here is the reason

Point A) The authors state

> Still, the linear fit to the directivity planes leads to a directivity of $\sim 0.35\%/nm \pm 0.0045\%/nm$ (95% confidence interval) as already mentioned in 'C. Calibration measurements' under the Methods section. The confidence interval statistically represents the deviation of individual points from the fitted plane. In other words, the linear fit is still very much appropriate even for distances larger than 100 nm.

This interpretation of confidence intervals misses the kind of errors that occur due to nonlinearity. I do not know exactly which method is being used to fit here, but I made a simple test using the nonlinear least-squares Marquardt-Levenberg algorithm fit provided by gnuplot. To this fit, I fed the same 21x21 data points within the $[-100,100] \times [100,100]$ nm range as used by the authors in Fig. 5(a). For these displacements, I defined the same response as in the paper (given by Eq. (7)) and made a small modification: for the first term, instead of the linear $0.33x - 0.02y - 1.75$, I used a cubic nonlinearity $0.2x + 2 \cdot 10^{-5}x^3 - 0.02y - 1.75$. Fitting with gnuplot gave the linear fit parameter as $0.33 \pm 0.0029\%/nm$, a lower error than found in the paper. And yet, when I back-calculated the positions using Eq. (8), I found errors as high as 17nm. This is much higher than $\lambda/300$ as claimed in the paper.

It is quite possible that the errors found by the authors are more noise-like and less systematic compared to my model system. Yet, to ensure that this is the case, it is necessary to show how well the piezo readout of the position matches with the back-calculated value from Eq. (8). A 2d map of the 21x21 points showing the respective position errors (piezo vs. back-calculated) would be completely convincing.

Point C) The authors state

> It is clear from Fig. 5c that the 11 sets of position data do not span a range of 250 nm owing to the effects of decreasing directivity (in a nonlinear fashion) beyond the central area.

This is not evident at all. The separation between points in Fig. 5c does not change monotonically. The way the points are spaced, it is likely to be due to error.

They also say

> Coming to the question of 'accuracy' vs. 'precision', we are convinced that using the standard deviation as an accuracy of the measurement (error bar) is even underestimating the capabilities of our current experimental setup in contrast to the Reviewer's claim of an increased accuracy. The directivity slopes of the fitted curve are between $0.33\%/nm$ and $0.37\%/nm$. Given with camera we can sense a change in the signal of way less than 1%, we are sure that the actual accuracy value would be even less than 5-6 nm.

I do not find this argument convincing. Directivity slopes only tell us about signal changes under large variations in position, not a lot about how well we can measure small displacements. In the end, an experimental situation is going to be such: A beam is incident on the system with no prior knowledge of where it is, and we need to find its position based on the signals from the six detectors. To claim 5-6 nm accuracy for such a detection, the authors need to show that their method is able to locate the position of a beam with that accuracy.

Reviewer #3 (Remarks to the Author):

The authors have satisfactorily revised the manuscript accordingly all my questions and remarks.

I find it now adequate for publication.

To Reviewer #1:

Comment 1 by Reviewer #1:

I thank the authors for the detailed reply and making most of the corrections accordingly.

Author's response:

We are very happy to read that we addressed most of the Reviewer's comments in a satisfactory and convincing manner.

Comment 2 by Reviewer #1:

However, I am unsatisfied with the author response to my queries about error analysis. Here is the reason.

> *The authors wrote: Still, the linear fit to the directivity planes leads to a directivity of $\sim 0.35\%/nm \pm 0.0045\%/nm$ (95% confidence interval) as already mentioned in 'C. Calibration measurements' under the Methods section. The confidence interval statistically represents the deviation of individual points from the fitted plane. In other words, the linear fit is still very much appropriate even for distances larger than 100 nm.* <

This interpretation of confidence intervals misses the kind of errors that occur due to nonlinearity. I do not know exactly which method is being used to fit here, but I made a simple test using the nonlinear least-squares Marquardt-Levenberg algorithm fit provided by gnuplot. To this fit, I fed the same 21x21 data points within the [-100,100] x [100,100] nm range as used by the authors in Fig. 5(a). For these displacements, I defined the same response as in the paper (given by Eq. (7)) and made a small modification: for the first term, instead of the linear $0.33x - 0.02y - 1.75$, I used a cubic nonlinearity $0.2x + 2 \cdot 10^{-5}x^3 - 0.02y - 1.75$. Fitting with gnuplot gave the linear fit parameter as $0.33 \pm 0.0029\%/nm$, a lower error than found in the paper. And yet, when I back-calculated the positions using Eq. (8), I found errors as high as 17nm. This is much higher than $\lambda/300$ as claimed in the paper.

It is quite possible that the errors found by the authors are more noise-like and less systematic compared to my model system. Yet, to ensure that this is the case, it is necessary to show how well the piezo readout of the position matches with the back-calculated value from Eq. (8). A 2d map of the 21x21 points showing the respective position errors (piezo vs. back-calculated) would be completely convincing.

Author's response:

It is very unfortunate that with our response we couldn't fully convince the Reviewer of the validity of our error analysis. We thank her/him again for very carefully assessing our scheme and for sharing corresponding thoughts and a simplified test calculation assuming a cubic nonlinearity. However, in our humble opinion, the generation of data by using a slight modification (resulting from a small additional nonlinear term) and then fitting the corresponding data using yet another nonlinear least square fit algorithm to compare with the actual data/fit does not provide a sound reasoning. We agree that doing a nonlinear fit would be ideal to cover even all negligible effects appearing near the outer rim of our localization window, but by staying inside this zone (100 nm x 100 nm) of linear dependence allows us to avoid this complications associated with it and helps us to keep the overall data analysis and position retrieval simple and scientifically correct. It is clear that the difference between some measured/retrieved position values and the real position might be as high as 17 nm. But the conclusion of $\lambda/300$ is not based on an individual measurement value. We performed a proper statistical error analysis (as mentioned in the main text and discussed in more detail below for Q4) on 64 measurement data points for each individual position set via the piezo-stage. In this context it is also important to note that piezo readouts do not capture the whole scenario. The aim is to localize the nanoparticle relative to the beam. Hence, the measured directivity signal may also change if the beam moves slightly while the particle might be standing still. The piezo readout is not capable of capturing all these vibrational and thermal motions of the system, which essentially cause the position fluctuations or errors. Hence, a 2D map of 21 x 21 points showing the piezo readout compared to the actual retrieved position of the particle with respect to the beam would not be too meaningful in the context of the assessment of localization errors.

Comment 3 by Reviewer #1:

> *The authors wrote: It is clear from Fig. 5c that the 11 sets of position data do not span a range of 250 nm owing to the effects of decreasing directivity (in a nonlinear fashion) beyond the central area.* <

This is not evident at all. The separation between points in Fig. 5c does not change monotonically. The way the points are spaced, it is likely to be due to error.

Author's response:

We thank the Reviewer also for this comment. Apparently, our response to the initial question did not unambiguously clarify this point. We are grateful for getting another chance to finally resolve this apparent misunderstanding. For our measurements, we try to not rely on the internal position information provided by the piezo stage. We send positioning data to the stage and measure what position the particle is sitting at based on our method. For a window of -100 nm ... +100 nm a linear dependence of directivity vs. position can be assumed. Outside this window, nonlinear terms kick in. This is why for the range of -125 nm ... + 125 nm, we do not measure 250 nm total range but less (the most outer positions are affected most by the nonlinear contributions to the dependence). However, the fluctuations we observe for all positions of the piezo-stage are a results of thermal fluctuations, noise, etc. The fact that the grey crosses (average positions) do not coincide with the set positions of $x=0; 25; 50\dots$ nm is a combined effect of non-perfect focal fields and the internal piezo-processing. This is also why a calibration is necessary. We hope this clarifies this point.

Comment 4 by Reviewer #1:

> *The authors wrote: Coming to the question of 'accuracy' vs. 'precision', we are convinced that using the standard deviation as an accuracy of the measurement (error bar) is even underestimating the capabilities of our current experimental setup in contrast to the Reviewer's claim of an increased accuracy. The directivity slopes of the fitted curve are between 0.33%/nm and 0.37%/nm. Given with camera we can sense a change in the signal of way less than 1%, we are sure that the actual accuracy value would be even less than 5-6 nm.* <

I do not find this argument convincing. Directivity slopes only tell us about signal changes under large variations in position, not a lot about how well we can measure small displacements. In the end, an experimental situation is going to be such: A beam is incident on the system with no prior knowledge of where it is, and we need to find its position based on the signals from the six detectors. To claim 5-6 nm accuracy for such a detection, the authors need to show that their method is able to locate the position of a beam with that accuracy.

Author's response:

This is still a question about accuracy and precision and the definition of their measures. Here in this manuscript we have used the 'standard deviation' σ as the measure of our accuracy. However, a 'standard error of the mean' is generally used to measure accuracy for such and other types of measurements [cite: Phys. Rev. Lett. 121, 193902 (2018)]. As the reviewer pointed out correctly, the directivity slopes only tell us about the signal change for large variations in position, but it would be also true for small displacements given there is a significant number of measurements taken from a statistical point of view. This number of measurements n plays a crucial role in the calculation of the accuracy. The standard error of the mean is defined as σ/\sqrt{n} . That is why we had mentioned that using standard deviation is an underestimation of the capabilities of our current experimental setup. For each position we have 64 measured data points, and using the aforementioned definition we arrive at an accuracy of less than 1 nm.

To prove this point with our experimental data, we show (in the Fig. below) the central position (red) and the two positions right next to the centre (blue in negative-x, and yellow in positive-x direction). The corresponding mean values are indicated by black circles. For all three positions, 64 measurement points are shown, leading to a standard deviation of 5-6 nm. The 'standard error of the mean' for this sampling data will be $6/\sqrt{64} < 1$ nm. In other words, the error in estimating the mean from these measurement points is less than 1 nm.

In addition, we would like to emphasize again here that we do not need to localize the beam with an accuracy of 5-6 nm but measure with our method the relative position of particle with respect to the beam. A piezo-stage would only measure the particle with respect to its internal coordinate frame not taking into account any relative movement of beam itself.

Fig.: Retrieved particle positions for the particle placed at three different positions set by the piezo-stage.

To Reviewer #3:

Comment by Reviewer #3:

The authors have satisfactorily revised the manuscript accordingly all my questions and remarks. I find it now adequate for publication.

Author's response:

We thank Reviewer #3 for his/her very valuable input, which helped us to improve our manuscript. We are very happy to read that we were able to address all comments in a satisfactory manner, and that the Reviewer now fully recommends publication of our work in Nat. Commun.

REVIEWERS' COMMENTS

Reviewer #1 (Remarks to the Author):

I am satisfied with the rebuttal of the authors. The manuscript can now be published.